# In Vitro Activity of Ceftolozane-Tazobactam and Other Antibiotics against *Pseudomonas aeruginosa* Infection-Isolates from an Academic Medical Center in Thailand

**DOI:** 10.3390/antibiotics11060732

**Published:** 2022-05-30

**Authors:** Woraphot Tantisiriwat, Jirawat Buppanharun, Chatchai Ekpanyaskul, Kwanchai Onruang, Thitiya Yungyuen, Pattarachai Kiratisin, Somchai Santiwatanakul

**Affiliations:** 1Department of Preventive Medicine, HRH Princess Maha Chakri Sirindhorn Medical Center, Faculty of Medicine, Srinakharinwirot University, Nakhon Nayok 26120, Thailand; jirawatbu2531@gmail.com (J.B.); dr_chatchai@hotmail.com (C.E.); 2Department of Pathology, HRH Princess Maha Chakri Sirindhorn Medical Center, Faculty of Medicine, Srinakharinwirot University, Nakhon Nayok 26120, Thailand; kwanchai@g.swu.ac.th (K.O.); somchaii@g.swu.ac.th (S.S.); 3Department of Microbiology, Faculty of Medicine Siriraj Hospital, Mahidol University, Bangkok 10700, Thailand; youngyuen_ja@yahoo.com (T.Y.); pattarachai.kir@mahidol.ac.th (P.K.)

**Keywords:** ceftolozane-tazobactam, *Pseudomonas aeruginosa*, in vitro susceptibility

## Abstract

(1) Background: Resistant *Pseudomonas aeruginosa* (PA) infections have limited treatment options. Data on the activity of ceftolozane-tazobactam (C-T) against PA in Thailand are limited. Objectives: The objective of this study was to identify the in vitro activity of C-T against general and resistant PA isolates from patients with real clinical infections from the HRH Princess Maha Chakri Sirindhorn Medical Center (MSMC) compared to other antibiotics and to study the resistant molecular patterns of those PA strains which were resistant to C-T. (2) Materials and Methods: This was an in vitro susceptibility study of 100 PA isolates plus an additional seven resistant PA isolates collected from MSMC patients. All PA isolates were tested with susceptibility broth (Sensititre™) and C-T minimal inhibitory concentration (MIC) test strips (Liofilchem, Roseto degli, Abruzzi, Italy). The C-T-resistant PA isolates were analyzed for six β-lactamase genes (*bla*_CTX-M_, *bla*_NDM_, *bla*_IMP_, *bla*_VIM_, *bla*_OXA-23_ and *bla*_OXA-48_) and the *mcr*-1 gene. (3) Results: A total of 100 PA isolates were collected between January 2020 and January 2021 and between February 2021 and September 2021 for the additional 7 resistant isolates. There were 18 resistant PA isolates (6 MDR, 11 XDR and 1 pan-drug resistant isolate). The overall susceptibility of the initial 100 PA isolates and the 18 resistant PA isolates was 94% and 44.5%, respectively, for C-T. The C-T susceptibility rates for isolates non-susceptible to ceftazidime, piperacillin-tazobactam, carbapenems and antipseudomonal β-lactams were 65.5%, 69.7%, 50% and 44.5%, respectively. Among the 10 isolates which were resistant to C-T, there were only 3 isolates found to have the resistant gene, which included 1 for *bla*_IMP_, 1 for *bla*_VIM_ and 1 for *bla*_NDM_. (4) Conclusions: Although C-T was the best susceptibility antibiotic overall for PA isolates and MDR PA isolates at the MSMC, most of the XDR PA isolates and the PDR PA isolate were not susceptible to C-T. The mechanisms for C-T resistance involved multiple factors including the presence of *bla*_IMP_, *bla*_VIM_ and *bla*_NDM_.

## 1. Introduction

*Pseudomonas aeruginosa* (PA) is among the most common bacteria, causing many types of hospital-associated infections [1]. The current Antibiotic Resistant Threats Report from the United States Centers of Disease Control and Prevention (CDC) in 2019 listed multidrug-resistant PA (MDR PA) among the serious threats [2,3]. The estimated number of cases in hospitalized patients in the U.S. was 32,600 and the estimated deaths were 2700 in 2017 [1].

MDR PA infections appeared to have an impact in Thailand [4,5,6]. Data from the National Nosocomial Resistance Surveillance, Thailand (NARST), showed an increased trend of carbapenem-resistant PA (CRPA) from approximately 10% during the years 2000–2005 to approximately 20% during 2016–2020 [4,5]. The review in 2016 (data from 10 institutions included 296 PA isolates) indicated that the prevalence of CRPA in Thailand was 28.7% [6].

The MDR PA infections were associated with poor outcomes [7,8,9]. There was limited availability of antibiotics for the treatment of difficult-to-treat PA-associated infections [10]. The current recommended antibiotics are ceftolozane-tazobactam, ceftazidime-avibactam, imipenem-relebactam, cefiderocol, colistin and aminoglycosides because of the in vitro susceptibility to these antibiotics [10]. Some of these antibiotics are novel, and there is limited information regarding their susceptibilities among PA isolates and the resistant PA isolates in Thailand.

The HRH Princess Maha Chakri Sirindhorn Medical Center (MSMC) is a 400-bed, tertiary-care university hospital in Nakhon Nayok around 80 km from Bangkok. PA is among the top five organisms causing hospital-associated infections at the MSMC. According to the previous MSMC antibiograms, resistant PA was found among 5–15% of all PA strains. The extremely drug-resistant (XDR) and pan-drug resistant (PDR) PA strains were resistant to most of the currently available antibiotics at the MSMC. The treatment options for these XDR and PDR PA infections are limited. The susceptibility data of the PA strains to ceftolozane-tazobactam is currently not available in the MSMC.

Primary objective: The authors would like to identify the in vitro activity of ceftolozane-tazobactam against 100 isolates of PA from patients with real clinical infections from the MSMC compared to other antibiotics.

Secondary objectives: The authors would also like to identify the in vitro activity of ceftolozane-tazobactam against the resistant PA isolates from patients with real clinical infections from the MSMC compared with the other antibiotics and study the resistant molecular patterns of those resistant PA strains that were resistant to ceftolozane-tazobactam.

## 2. Materials and Methods

Study design: This was a laboratory experiment study of the in vitro susceptibility of PA isolates collected from MSMC patients with real clinical infections. The PA study isolates included only the first recovered isolate from each eligible patient. Isolates which were cultured from colonization or contamination were excluded from the present study. Source clinical data were collected, along with the previous antibiotics use information. Isolates from the same sources with the same susceptibility pattern were also discarded.

The total number of initial isolates to be tested was 100, which were analyzed to identify MDR, XDR and PDR PA isolates. The estimated number of resistant PA isolates was 15–20, and more would be collected if the initial number of resistant PA isolates did not reach this estimation from the first 100 PA strains collected. If the number of resistant PA strains needed to be more than the initial resistant isolates collection, additional resistant PA strains were added to reach the maximum amount of resistant isolates which could be collected.

Inclusion criteria: All clinical culture specimens from the MSMC patients which grew PA were collected from the MSMC microbiological laboratory from 13 January 2020 onwards until PA isolate collection was complete. The initial MSMC susceptibility test was the VITEK^®^ 2 compact (bioMérieux, Craponne, France). This susceptibility system was used to test all PA stains for genus, species and susceptibilities. Following that, the organisms were stored in Skim Milk (BD BBL™/Difco^™^/Additives/Ingredients/Reagents: Skim Milk by Becton Dickinson Life Sciences, Franklin Lakes, NJ, USA) and frozen at −70 °C. The source data were reviewed for real clinical infections by the present study’s clinical team (the principal investigator, W.T., and the sub-investigator, J.B., whose clinical judgement had to be the same for the infection to be considered real). The sources which were confirmed as clinical infections were identified to the microbiological laboratory staff, who then included those PA isolates in the present study.

Exclusion criteria: PA isolates from sources which were subject to contamination or colonization were reported to the microbiological laboratory staff and discarded. PA isolates with the same microbiological sensitivity from a patient would be included only once in the study period to avoid duplication.

Definitions: The present study classified PA strains as MDR PA when the strain showed resistance to at least 1 antibiotic from the 3 available antibiotic classes in the present study antibiotic susceptibility report [11]. The available antibiotic classes in the present study consisted of antipseudomonal carbapenems (imipenem and meropenem), aminoglycosides (gentamicin, tobramycin and amikacin), antipseudomonal fluoroquinolone (ciprofloxacin), antipseudomonal cephalosporin (ceftazidime), antipseudomonal penicillin-beta-lactamase inhibitor (piperacillin-tazobactam), colistin and monobactam (aztreonam). For the XDR PA, there were only 1 or 2 antibiotic classes which were susceptible [11]. For PDR PA, there was no susceptible antibiotic in the susceptibility panel [11]. Ceftolozane-tazobactam was not included in the definition criteria.

Data Collection: After the PA isolates were included in the present study, the present study’s clinical information was collected by reviewing the patients’ medical records and electronic medical records. The data included patient identification, the infection diagnosis and the patients’ antibiotic use before the diagnosis of PA infection.

PA isolates were stored in a −70 °C freezer prior to the study. Before testing, the isolates were subcultured twice on 5% sheep blood agar and incubated at 35–37 °C for 18–24 h. All PA isolates were tested using the present study susceptibility broth (Sensititre Gram-negative plate format; plate code: DKMGN by Thermo Scientific, Lenexa, KS, USA) and ceftolozane-tazobactam minimal inhibitory concentration (MIC) test strips (Liofilchem, Roseto degli, Abruzzi, Italy). All PA isolates were adjusted for turbidity bacterial concentration at 0.5 McFarland Standard before MIC testing. All PA isolates were tested with the Gram-negative MIC plates of Sensititre DKMGN following the standard protocol of Thermo Scientific Sensititre Susceptibility and Identification System and incubated at 35–37 °C ambient air for 16–18 h. The ceftolozane-tazobactam MIC strips were tested with the PA isolates on the Mueller–Hinton agar plate and incubated at 35–37 °C ambient air for 16–18 h. The MIC interpretation was conducted by using the MIC Breakpoints for PA [12]. The interpretation of the MIC was: susceptible (≤4), intermediate (=8) and resistant (≥16) [12]. PA ATCC 27853 and *Escherichia coli* ATCC 25922 were used for internal quality control. The susceptibility data and MIC value of each PA isolate were recorded.

PA isolates which were resistant to ceftolozane-tazobactam were sent to the microbiology department of the Faculty of Medicine Siriraj Hospital for molecular analysis. To detect the presence of selected resistance genes, bacterial DNA was extracted from the overnight culture using the boiling method. A bacterial suspension in Tris-EDTA buffer was heated at 95 °C for 15 min and was then centrifuged at 13,500 rpm for 5 min. The supernatant was collected for a Polymerase Chain Reaction (PCR) study to detect six β-lactamase genes (*bla*_CTX-M_, *bla*_NDM_, *bla*_IMP_, *bla*_VIM_, *bla*_OXA-23_ and *bla*_OXA-48_) and a colistin-related resistance gene (*mcr*-1). The PCR primers and conditions were set according to previously published references (Table 1) [13,14,15,16,17,18]. The 16S rDNA was used as internal control for the PCR [19].

Statistical and data analysis: All data analyses were performed using the Statistical Package for Social Sciences version 22.0 (IBM Corp, Armonk, NY, USA). The descriptive statistics for the clinical data were presented as frequency and proportion in percentage or mean standard deviation, depending on the characteristics of the data. The infections were classified by the number of each systematic infection. Antibiotic use before infection was classified by the duration of use and antibiotic class before PA infection. The susceptibility data of each antibiotic were described using numbers and percentages. The MIC values were presented as numbers. In terms of inferential statistics, the estimation of drug susceptibility against PA is presented as a sensitivity rate with 95% confidence intervals. Comparison of susceptibility between ceftolozane-tazobactam and the other antibiotics utilized the McNemar’s test. The comparison between previous antipseudomonal antibiotic(s) exposure and PA resistance utilized logistic regression and is presented as an odds ratio with 95% confidence intervals. The significance level was considered at a *p*-value of less than 0.05.

## 3. Results

A total of 100 PA isolates which caused clinical infections were collected between January 2020 and January 2021. There was a total of 11 resistant PA isolates among the initial 100 PA isolates. Additional resistant PA isolates were collected to increase the number of resistant PA isolates between February 2021 and September 2021. There was a total of 7 resistant PA isolates which had been collected during this period. The study flowsheet is presented in Figure 1.

The initial 100 PA isolates were from 64 males and 36 females. There were 45 lower respiratory tract infections, 36 blood stream infections, 11 urinary tract infections, 9 intra-abdominal infections and 5 others (2 sinusitis, 2 otitis and 1 infected neck mass). The additional 7 resistant PA isolates were from 6 males and 1 female. There were 3 lower respiratory tract infections, 2 intra-abdominal infections, 1 urinary tract infection and 1 skin and soft tissue infection.

The overall susceptibility of the initial 100 PA isolates is presented in Table 2.

The gradient strip MIC data for ceftolozane-tazobactam among these initial 100 isolates were from 0.19 µg/mL to >256 µg/mL (1 at 0.19 µg/mL; 2 at 0.25 µg/mL; 19 at 0.38 µg/mL; 35 at 0.5 µg/mL; 24 at 0.75 µg/mL; 5 at 1 µg/mL; 1 at 1.5 µg/mL; 4 at 2 µg/mL; 3 at 3 µg/mL; and 6 at >256 µg/mL). The gradient strip MIC data for ceftolozane-tazobactam among the 7 additional resistant PA isolates were from 1 µg/mL to >256 µg/mL (3 at 1 µg/mL; 1 at 2 µg/mL; and 4 at >256 µg/mL). The gradient strip MIC results were correlated with the broth microdilution results. The gradient strip MIC values for ceftolozane-tazobactam among the 107 isolates are presented in Figure 2.

The responses of these antibiotics were compared to the responses of ceftolozane-tazobactam and another antibiotic individually using McNemar’s test. While the response rates which were compared between antibiotics were between susceptible versus non-susceptible (intermediate and resistance), the comparison between ceftolozane-tazobactam and colistin was between susceptible and intermediate versus resistance. There were statistically significant responses for ceftolozane-tazobactam (in favor of ceftolozane-tazobactam) compared to meropenem, imipenem, ciprofloxacin, ceftazidime, aztreonam and piperacillin-tazobactam. There were no statistically significant responses between ceftolozane-tazobactam and colistin, amikacin, gentamicin and tobramycin, as described in Table 2.

Among the 18 resistant PA isolates, 1 PDR isolate was resistant to all 7 of the present study antibiotic classes. Among the 11 XDR isolates, 8 isolates were resistant to 6 of the present study antibiotic classes and 3 isolates were resistant to 5 of the present study antibiotic classes. Of the 6 MDR isolates, 4 were resistant to 4 of the present study antibiotic classes and 2 were resistant to 3 of the present study antibiotic classes. This information is presented in Table 3.

The overall susceptibility of these 18 resistant PA isolates is presented in Table 4.

The overall susceptibilities of the 6 MDR PA isolates were 100% for ceftolozane-tazobactam and tobramycin, 83.3% for gentamicin, 66.7% for ciprofloxacin and amikacin, 33.3% for meropenem, 16.7% for imipenem and aztreonam and 0% for ceftazidime and piperacillin-tazobactam. There was a 100% intermediate response to colistin. The MIC data for ceftolozane-tazobactam among these isolates ranged from 0.75 µg/mL to 3 µg/mL (1 for 0.75 µg/mL; 3 for 1 µg/mL; 1 for 2 µg/mL; and 1 for 3 µg/mL).

The overall susceptibilities of the 11 XDR PA isolates were 18.2% for ceftolozane-tazobactam, tobramycin, gentamicin and amikacin; 9.1% for ciprofloxacin; and 0% for meropenem, imipenem, ceftazidime, aztreonam and piperacillin-tazobactam. There was an 81.8% intermediate response to colistin. The MIC data for ceftolozane-tazobactam among these isolates ranged from 0.75 µg/mL to > 256 µg/mL (1 for 0.75 µg/mL; 1 for 1 µg/mL; 9 for > 256 µg/mL).

For the single PDR PA isolate, there were no susceptible antibiotics available in this present study. The PDR PA isolate was also resistant to colistin. The MIC data for ceftolozane-tazobactam for the PDR PA isolate was >256 µg/mL.

The susceptibilities of ceftolozane-tazobactam for isolates non-susceptible to various antibiotics are described in Table 5. The antipseudomonal β-lactams included ceftazidime, piperacillin-tazobactam, carbapenems and aztreonam.

According to the history of antibiotic exposure before the PA infections issue, there was a total of 13 patients (in the 18 patients with resistant PA isolates group) who received anti-pseudomonal antibiotic(s) within three months prior to the onset of PA infection. In the 89 patients without the resistant PA isolates group, there were only 15 patients who received anti-pseudomonal antibiotic(s) within three months prior to the onset of the PA infection. The case–control analysis found the odds ratio of the previous antipseudomonal antibiotic(s) exposure related with resistant PA isolates to be 12.83 (95% CI = 3.98–41.38, *p*-value < 0.001), as presented in Table 6.

Among the 18 resistant PA isolates, there were 10 isolates which were resistant to ceftolozane-tazobactam. These isolates were analyzed for the presence of selected resistance genes, which included the *bla*_CTX-M_, *bla*_NDM_, *bla*_IMP_, *bla*_VIM_, *bla*_OXA-23_, *bla*_OXA-48_ and *mcr*-1 genes. Only 3 isolates were found to have resistant genes which included 1 isolate with *bla*_IMP_, 1 isolate with *bla*_VIM_ and 1 isolate with *bla*_NDM_. All isolates were classified as XDR. Two isolates were intermediate to colistin and resistant to the rest of the antibiotics (isolates with *bla*_VIM_ and *bla*_NDM_) and the isolate with *bla*_IMP_ was intermediate to colistin, piperacillin-tazobactam and amikacin but resistant to the rest of the antibiotics. The MIC data for ceftolozane-tazobactam among these isolates were >256 µg/mL.

## 4. Discussion

PA is one of the most common pathogens causing hospital-associated infections [1]. Current data from the global Study for Monitoring Antimicrobial Resistance Trends (SMART) surveillance program from 2017–2019 ranked PA third among Gram-negative isolates from intensive care patients with lower respiratory tract infections in Thailand [20].

Ceftolozane-tazobactam is an anti-pseudomonal cephalosporin combined with a beta-lactamase inhibitor which was licensed for the treatment of lower respiratory tract infections, urinary tract infections and intra-abdominal infections in Thailand in 2017 [21]. Lob S.H. et al., from the SMART study, recently reported the overall susceptibility of 761 PA isolates to ceftolozane-tazobactam from ‘seven Asian countries to be 85.9%, while the overall susceptibility of 126 PA isolates from Thailand to ceftolozane-tazobactam was 74.6% [20]. The susceptibility of 186 resistant PA (resistant to meropenem and piperacillin-tazobactam) isolates from 7 Asian countries to ceftolozane-tazobactam was 51.1%, while the susceptibility of 37 resistant PA isolates from Thailand to ceftolozane-tazobactam was only 18.9% [20].

Ceftolozane-tazobactam had a more statistically significant susceptibility percentage in comparison to meropenem, imipenem, ciprofloxacin, ceftazidime, aztreonam and piperacillin-tazobactam against PA isolates but not statistically significant in comparison to colistin and the aminoglycosides in the present study. Ceftolozane-tazobactam is currently not available at the MSMC. This reason could explain the very high susceptibility percentage of ceftolozane-tazobactam of the PA isolates in the present study compared with the recent data reported by SMART Thailand.

The 18 resistant PA isolates in the present study can be separated into two groups. The first group is the MDR isolates which are resistant to at least 3–4 antibiotic classes. All MDR isolates were susceptible to ceftolozane-tazobactam with an MIC distribution ranging from 0.75 µg/mL to 3 µg/mL. The other antibiotics which were considered were colistin (I = 100%) and ciprofloxacin (66.7% in cases of proven sensitivity). The other antibiotics were not good for empirical treatment if the MDR PA isolates were in the differential diagnosis.

The second group is the XDR and PDR isolates. For these extremely resistant PA infections, ceftolozane-tazobactam would not be good for empirical coverage. There were 2 isolates from the present study which were sensitive to ceftolozane-tazobactam. Both were sensitive to the aminoglycosides and were intermediate to colistin, and 1 isolate was also sensitive to ciprofloxacin. Although there were not many antibiotics available for these infections, the mainstem of the treatment was to find susceptible antibiotics to be used. All XDR and PDR isolates should be tested thoroughly to find the available antibiotics for the best available treatment.

Among the carbapenem non-susceptible PA isolates, there were 80% with intermediate response to colistin, 50% with sensitive response to ceftolozane-tazobactam and 40–45% with sensitive response to the aminoglycosides. Ceftolozane-tazobactam should be considered for the treatment of carbapenem-resistant PA infections at the MSMC if its sensitivity is proven.

Ceftolozane-tazobactam was more than 60% susceptible to isolates non-susceptible to ceftazidime and piperacillin-tazobactam. It appeared to be a good choice for the treatment of infections with these pathogens in the MSMC.

One of the risk factors for resistant PA infections identified in the present study was the previous use of antipseudomonal antibiotic(s). Resistant PA infections were associated with the use of antipseudomonal antibiotic(s) within the past three months with an odds ratio of 12.83. In the only PDR PA isolate found in the present study, there was an association with multiple antipseudomonal antibiotics involvement before the identification of this PDR PA isolate. Most of the present study patients without previous antipseudomonal antibiotic(s) exposure did not have resistant PA infections.

Only 3 isolates were identified as having resistant genes, including *bla*_IMP_, *bla*_VIM_ and *bla*_NDM_, among a total of 10 ceftolozane-tazobactam–resistant PA isolates. Khuntayaporn, P. et al. reported that another gene consideration for PA resistance in Thailand could be from major sequence type (ST) PA isolates (ST 235, ST 964 and ST 111) [22]. The ST types of these PA isolates were not identified in the present study. This finding indicates that the mechanisms for resistant PA infections at the MSMC could involve multiple factors. Other resistant mechanisms could be the ST types of the resistant gene [22] or other resistant mechanisms from efflux pumps and porin changes [23].

The present study had some limitations. The nature of a single-center study and the one-year period to collect 100 isolates likely represented few clones, and the results may not be generalizable.

## 5. Conclusions

Although ceftolozane-tazobactam was generally the best susceptibility antibiotic for PA isolates and the MDR PA isolates at the MSMC, most of the XDR PA isolates and the PDR PA isolate were not susceptible to ceftolozane-tazobactam. The mechanisms for resistance involved multiple factors, including the presence of *bla*_IMP_, *bla*_VIM_ and *bla*_NDM_.

## Figures and Tables

**Figure 1 antibiotics-11-00732-f001:**
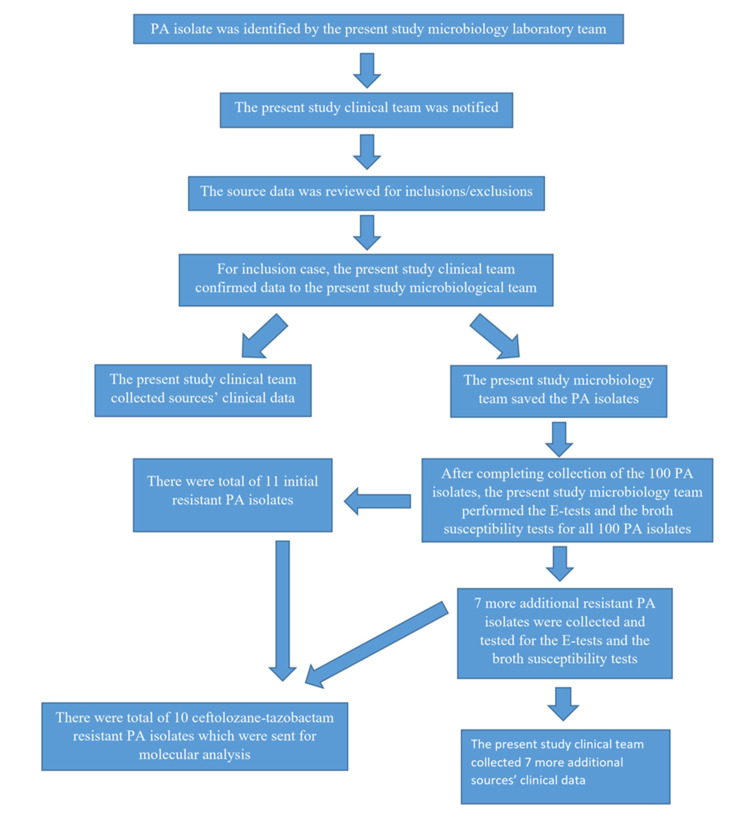
Study flowsheet.

**Figure 2 antibiotics-11-00732-f002:**
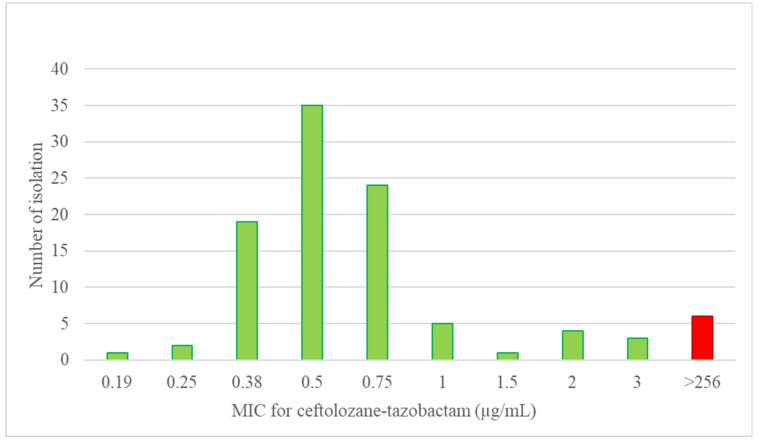
The gradient strip MIC distribution for ceftolozane-tazobactam among the 107 present study PA isolates.

**Table 1 antibiotics-11-00732-t001:** PCR primers.

Target Gene	Primer	References
*bla* _CTX-M_	(F) 5′-GCGATGTGCAGCACCAGTAA-3′ (R) 5′-GGTTGAGGCTGGGTGAAGTA-3′	[13]
*bla* _NDM_	(F) 5′-GGTTTGGCGATCTGGTTTTC-3′ (R) 5′-CGGAATGGCTCATCACGATC-3′	[14]
*bla* _IMP_	(F) 5′-GGAATAGAGTGGCTTAAYTCTC-3′ (R) 5′-CCAAACYACTASGTTATCT-3′	[15]
*bla* _VIM_	(F) 5′-GATGGTGTTTGGTCGCATA-3′ (R) 5′-CGAATGCGCAGCACCAG-3′	[16]
*bla* _OXA-23_	(F) 5′-GATCGGATTGGAGAACCAGA-3′ (R) 5′-ATTTCTGACCGCATTTCCAT-3′	[17]
*bla* _OXA-48_	(F) 5′-GCGTGGTTAAGGATGAACAC-3′ (R) 5′-CATCAAGTTCAACCCAACCG-3′	[14]
*mcr*-1	(F) 5′-CGGTCAGTCCGTTTGTTC-3′ (R) 5′-CTTGGTCGGTCTGTAGGG-3′	[18]

**Table 2 antibiotics-11-00732-t002:** The overall susceptibility of the initial 100 PA isolates to each antibiotic.

Antimicrobial Agent	Susceptibility (*n*)			*p*-Value Compared to C-T
	S	I	R	
Ceftazidime	78	7	15	<0.001
Piperacillin-tazobactam	74	11	15	<0.001
Aztreonam	74	10	16	<0.001
Meropenem	83	6	11	0.001
Imipenem	84	3	13	0.002
Gentamicin	91	3	6	0.25
Amikacin	90	4	6	0.125
Tobramycin	89	3	8	0.063
Ciprofloxacin	84	5	11	0.002
Colistin	-	96	4	0.687
Ceftolozane-tazobactam	94	-	6	

**Table 3 antibiotics-11-00732-t003:** The characteristics of the 18 resistant PA isolates and the MIC of ceftolozane-tazobactam for each isolate. * Isolate with *bla*_IMP_, ** Isolate with *bla*_NDM_ and *** Isolate with *bla*_VIM_.

Isolation Number	MIC (µg/mL)	Antibiotic Resistance	Type
1	>256	All antibiotics	PDR
8	0.75	CIP, CAZ, TZP, ATM, IPM	XDR
11	1	CIP, ATM, MEM	MDR
27	1	CAZ, TZP, ATM, IPM, CST	XDR
30	3	CAZ, TZP, ATM	MDR
31	0.75	CAZ, TZP, ATM, MEM, IPM	MDR
37	>256	CIP, CAZ, ATM, MEM, IPM, GEN, TOB, CST, C-T	XDR
52 *	>256	CIP, CAZ, ATM, MEM, IPM, GEN, TOB, C-T	XDR
64	>256	CIP, CAZ, TZP, ATM, MEM, IPM, AMK, GEN, TOB, C-T	XDR
77 **	>256	CIP, CAZ, TZP, ATM, MEM, IPM, AMK, GEN, TOB, C-T	XDR
92	>256	CIP, CAZ, TZP, ATM, MEM, IPM, AMK, GEN, TOB, C-T	XDR
101	1	CIP, CAZ, TZP, IPM	MDR
102	>256	CIP, CAZ, TZP, ATM, MEM, IPM, AMK, GEN, TOB, C-T	XDR
103 ***	>256	CIP, CAZ, TZP, ATM, MEM, IPM, AMK, GEN, TOB, C-T	XDR
104	>256	CIP, CAZ, TZP, ATM, MEM, IPM, AMK, GEN, TOB, C-T	XDR
105	2	CAZ, TZP, ATM, MEM, IPM	MDR
106	>256	CIP, CAZ, TZP, ATM, MEM, IPM, AMK, GEN, TOB, C-T	XDR
108	1	CAZ, TZP, ATM, MEM, IPM	MDR

**Table 4 antibiotics-11-00732-t004:** The overall susceptibility of the 18 resistant (MDR, XDR or PDR) PA isolates to each antibiotic.

Antimicrobial Agent	Susceptibility (*n*)			*p*-Value Compared to C-T
	S	I	R	
Ceftazidime	-	1	17	0.008
Piperacillin-tazobactam	-	3	15	0.008
Aztreonam	1	-	17	0.016
Meropenem	2	2	14	0.031
Imipenem	1	1	16	0.016
Gentamicin	7	1	10	1.0
Amikacin	6	4	8	0.5
Tobramycin	8	-	10	1.0
Ciprofloxacin	5	-	13	0.25
Colistin	-	15	3	0.039
Ceftolozane-tazobactam	8	-	10	

**Table 5 antibiotics-11-00732-t005:** The susceptibility of ceftolozane-tazobactam for isolates non-susceptible to various antibiotics.

Non-Susceptible Antibiotic (A)	C-T Susceptibility (B)
	B/A (%)
Ceftazidime	19/29 (65.5)
Piperacillin-tazobactam	23/33 (69.7)
Aztreonam	23/33 (69.7)
Carbapenems	10/20 (50.0)
Antipseudomonal β-lactams	8/18 (44.5)

**Table 6 antibiotics-11-00732-t006:** The case–control analysis for the resistant PA infections and the previous antipseudomonal antibiotic(s) exposure.

Previous Antipseudomonal Antibiotic(s) Exposure	All Isolations		Total
	Resistance (MDR, XDR or PDR)	Non-Resistance	
Yes	13 (46.4%)	15 (53.6%)	28 (100%)
No	5 (6.3%)	74 (93.7%)	79 (100%)
Total	18	89	*n* = 107

## Data Availability

The data that support the finding of this present study are available on request from the corresponding author.

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
