# Peer review of "In Vitro Activity of Ceftolozane-Tazobactam and Other Antibiotics against Pseudomonas aeruginosa Infection-Isolates from an Academic Medical Center in Thailand"

_antibiotics, 2022, doi:10.3390/antibiotics11060732_

Round 1
Reviewer 1 Report
Overall report: I like the topic about looking at antimicrobial resistance(AMR) in resource limited settings. AMR is a serious problem and needs global intervention . I feel studies like these will help create more awareness and traction.
Flow: Flow of the manuscript and its interest to the reader is satsifactory
Language: satisfactory
Statistical methods: Appropriate
Content validity: In the age of indiscrimate use of antimicrobials and subsequent AMR, I feel this is a good study
Author Response
Thank you very much for the review and comment.
Kob Kun Krub
Woraphot Tantisiriwat, MD
Reviewer 2 Report
Thank you for the article. The article can be accepted in its current form, but for my own curiosity, would you mind elaborating on the fact, of how the clinical data detected the contamination. Otherwise, I have no other suggestion.
Author Response
Thank you very much for the review. We used the clinical information together with the laboratory/imaging results and the antibiotic use for the situations to decide the contaminations or infections.
Sincerely,
Woraphot Tantisiriwat, MD
Woraphot
Reviewer 3 Report
Overall an interesting paper on the antibiotics targeting PA in vitro in Thailand. I have attached a PDF with changes required to spelling etc. before acceptance for publication.

Author Response
Thank you very much for the review and correction for the manuscript.
I have corrected the manuscipt as suggestion (appeared in red color)
Kob Kun Krub,
Woraphot Tantisiriwat, MD